# Spatiotemporal Associations between Local Safety Level Index and COVID-19 Infection Risks across Capital Regions in South Korea

**DOI:** 10.3390/ijerph19020824

**Published:** 2022-01-12

**Authors:** Youngbin Lym, Hyobin Lym, Keekwang Kim, Ki-Jung Kim

**Affiliations:** 1Research Institute of Natural Sciences, Chungnam National University, Daejeon 34134, Korea; youngbin.lym@gmail.com; 2Korea Rural Economic Institute, Naju-si 58321, Korea; hyobin@krei.re.kr; 3Department of Biochemistry, Chungnam National University, Daejeon 34134, Korea; 4Department of Smart Car Engineering, Doowon Technical University, Anseong 10838, Korea

**Keywords:** Spatiotemporal Bayesian, COVID-19, infection risks, local safety level index, Seoul capital area, unobserved heterogeneity

## Abstract

This study aims to provide an improved understanding of the local-level spatiotemporal evolution of COVID-19 spread across capital regions of South Korea during the second and third waves of the pandemic (August 2020~June 2021). To explain transmission, we rely upon the local safety level indices along with latent influences from the spatial alignment of municipalities and their serial (temporal) correlation. Utilizing a flexible hierarchical Bayesian model as an analytic operational framework, we exploit the modified BYM (BYM2) model with the Penalized Complexity (PC) priors to account for latent effects (unobserved heterogeneity). The outcome reveals that a municipality with higher population density is likely to have an elevated infection risk, whereas one with good preparedness for infectious disease tends to have a reduction in risk. Furthermore, we identify that including spatial and temporal correlations into the modeling framework significantly improves the performance and explanatory power, justifying our adoption of latent effects. Based on these findings, we present the dynamic evolution of COVID-19 across the Seoul Capital Area (SCA), which helps us verify unique patterns of disease spread as well as regions of elevated risk for further policy intervention and for supporting informed decision making for responding to infectious diseases.

## 1. Introduction

The severe acute respiratory syndrome-2 (SARS-CoV-2, also known as COVID-19) was first reported in Wuhan, China in December 2019 and has continued to spread across the globe [1]. Consequently, the WHO characterized COVID-19 as a global pandemic on 11 March 2020 [2]. As of 31 October 2021, more than 242 million COVID-19 cases were confirmed worldwide, with more than 4.9 million deaths attributed to the infectious disease [3]. Meanwhile, in South Korea there have been a total of 364,700 confirmed cases (704 per 100,000 population) of COVID-19 reported, with 2849 deaths (5.50 per 100,000 population) as of 31 October 2021 [4]. 

The first confirmed case of COVID-19 in Korea was an inbound Chinese female traveler from Wuhan, China, on 19 January 2020 [5]. Since then, the pandemic has hit the entire nation severely with a series of unexpected surges of events characterized by four heterogeneous waves. South Korea is currently going through its fourth wave of the COVID-19 pandemic, in which the number of daily confirmed cases has appeared to be more than 1000 for 100 consecutive days since early July 2021 [6]. Additionally, the first, second, and third waves occurred in February, August, and November 2020, respectively. In response to each wave, the government of South Korea introduced and implemented several countermeasures against COVID-19, including a series of social distancing schemes, mandatory mask wearing, and active tracking of the infected, to decrease its spread without a nationwide lockdown. To this end, we have witnessed that through the end of June 2021, the pattern of COVID-19 infections in South Korea rather differs from that of other countries; this is due to the effectiveness of countermeasures and people’s compliance with the policy measures. We are also aware that there have been varied outcomes in terms of COVID-19 across nations as each country has responded differently to the disease. With this in mind, it is worth investigating how the virus spreads and evolves across regions over time. 

According to the WHO (2021) [7], the coronavirus spreads primarily between people in close contact with each other, typically within a short distance (i.e.., about a meter). People can also become infected when aerosols or droplets containing the virus are inhaled or come in direct contact with their eyes, nose, or mouth. Additionally, a study suggests the possibility of airborne transmission of the coronavirus in crowded, closed, and poorly ventilated public environments [8]. Numerous attempts have been made to find various influential factors, such as built environments, environmental conditions, socio-demographic features, and latent effects including spatial dependency and temporal/serial correlation that affect the transmission of COVID-19 [9,10,11,12,13].

Meanwhile, several scholars have investigated the spread of COVID-19 in Korea from various perspectives. Kim and Castro [14] conducted a retrospective space-time scan statistic to identify 12 spatiotemporal clusters of early stages of COVID-19 in South Korea. Shim et al. [15] estimated the spatial variability in the reproduction number and doubling time of COVID-19 in South Korea based on daily cases collected from January to July 2020. In a similar vein, Lee et al. [16] showed the spread pattern via a logistic growth model along with the calculation of the daily reproduction number and the examination of the fatality pattern at an early stage. Lym and Kim [17] verified that PM2.5 concentration and temperature have significant influences on the daily COVID-19 transmission risks across 25 wards in Seoul. 

From a different perspective, considering multi-faceted features of COVID-19 transmission mechanisms, this study attempts to examine the association between social features/environments and spatiotemporal evolution of the COVID-19 pandemic at the local municipality level of the Seoul Capital Area (SCA) in South Korea. In particular, ‘local safety index’ is used as a sociodemographic factor affecting the spread of COVID-19 in this paper. Here, the ‘local safety index’ is an evaluation of the safety level of local governments by the Ministry of the Interior and Safety (MOIS) into five grades in six areas (traffic accidents, fires, crimes, living safety, suicide, and infectious disease), with the purpose of reducing the number of fatal accidents and strengthening safety capabilities [18]. Since all indices were standardized in terms of population or administrative district area, we regard them as representative of regional sociodemographic features. Furthermore, our research focuses on clarifying the role of latent influences that originate from neighboring locations (spatial dependence characterized by adjacency of municipalities) and temporal correlation. We employ a flexible Bayesian hierarchical approach to account for these random effects. To this end, we would like to answer the following: (1) whether the local safety level (or regional preparedness for external threats such as COVID-19) contributes to reducing the risk of COVID-19 transmission; (2) if the incorporation of unobserved heterogeneity stemming from spatial alignments and temporal dependency enhance the explanatory power of the model; (3) where the hot spots for further policy consideration are. We believe that we can contribute to improving our understanding of COVID-19 evolution over space and time for the capital regions of South Korea and support evidence-informed decision making to mitigate its spread.

## 2. Data and Methodology

### 2.1. Background

As shown in Figure 1, South Korea has experienced four heterogeneous waves of COVID-19 since its outbreak in January 2020. The first wave broke out in the southeastern city of Daegu and continued until early May. At that time, the number of daily new coronavirus cases reached 813 on 29 February 2020. 

The second wave took place in mid-August 2020. Cases were mostly associated with large gatherings at religious facilities and various multi-use facilities during this period. On 26 August 2020, the daily number of new confirmed cases was 441 [19]. 

Meanwhile, the trend has changed since the third wave, which started on 13 November 2020, as the number of sporadic infections and hidden infections with unknown routes of infection increased significantly. The daily maximum has reached 1237 on 24 December 2020 amidst the third wave [20]. 

Finally, the fourth wave, though not included in this research, started in early July 2021. So far, more than 1000 confirmed cases per day have been identified on an ongoing basis [21].

### 2.2. Data Description

#### 2.2.1. Study Region

According to KDCA [22], 68% of the total confirmed cases occurred in the densely populated SCA. Additionally, as of 5 November 2021, Seoul had the highest incidence rate of COVID-19 (1270 per 100,000 people), followed by Gyeonggi-do (829), and Daegu (718), which is the epicenter of the first wave. The first wave was triggered by a church in Daegu; appropriate containment measures prevented spillover, and the situation stabilized quickly. However, the pandemic spread in a different way within the SCA, making its control challenging [23]. This motivates us to focus on capital regions rather than other places, and we attempt to explore the dynamic evolution of COVID-19 over space across the SCA (Figure 2).

This study considers 77 districts (i.e., si/gun/gu) in the SCA, which includes Seoul, Gyeonggi province, and Incheon, where 50% of the total population of Korea resides. For each district (municipality), we collected daily (time) series of confirmed cases of COVID-19 from 24 January 2020 (the first day that COVID-19 confirmed cases were observed in the SCA) to 30 June 2021. Subsequently, a series of validation and data manipulation procedures was conducted. We specifically excluded cases due to inbound travelers from foreign countries because not only could those records confound the outcomes of the analysis, but they were also not fully available (unidentifiable) in earlier periods. In addition, we frequently visited 77 municipal websites to validate and improve the accuracy of our dataset due to the readjustment of COVID-19 records (e.g., reporting delays or errors). This allowed us to ensure that we had detailed and precise daily COVID-19 infection data. The daily counts of COVID-19 confirmed cases were then converted to monthly aggregates for subsequent analysis. We obtained geospatial information data from OpenMarket at the National Spatial Data Infrastructure Portal [24]. In preparation for the formal modeling process, we related spatial data to monthly confirmed cases as well as sociodemographic features using a unique identifier of each district (i.e., a five-digit code for an administrative unit). Districts that shared a common border (boundary) were regarded as neighbors in this study.

#### 2.2.2. Distribution of COVID-19 Confirmed Cases 

This study mainly considers the second and third waves of the COVID-19 pandemic for the SCA. As described in Figure 2, the SCA consists of three provinces with 77 municipalities. To identify overall regional disparities in COVID-19 infections among three provinces in the SCA, we present Figure 3, which shows the frequency distribution of monthly aggregated counts per 100,000 population. Clearly the distribution of COVID-19 cases per 100,000 people varies across provinces, with average values of 44.1 in Seoul, 17.2 in Incheon, and 32.7 in Gyeonggi. These numbers, however, do not fully depict local district-level infection risk in the SCA, leading us to develop in-depth investigation of COVID-19 spread across the local municipalities over 11 months (August 2020~June 2021). 

#### 2.2.3. Data Adopted in This Study

For 77 local municipalities in the SCA, this study attempts to examine the association between social features/environments and spatiotemporal dynamics of COVID-19 diffusion. We select a local municipality/district (i.e., si/gun/gu, an administrative unit) as a unit of analysis for data collection and manipulation. For each municipality, a time-series of daily confirmed cases of COVID-19 was collected and then converted to monthly aggregates over the study period (from 1 August 2020 to 30 June 2021). Covariates specific to demographic features of the regions were collected from the Korean Statistical Information Service [25] as shown in Table 1. In addition, we consider the local safety level index in 2019, provided by the Ministry of the Interior and Safety of South Korea (MOIS) [26]. The indices consist of six key areas including traffic accidents, fires, crime, living safety, and infectious diseases calculated by the weighted sum of the harm index, cause index, and mitigation index. There are five levels depicting local level risk factors and conditions (i.e., Level 1: Best, Level5: Worst). One can refer to https://www.mois.go.kr/eng/sub/a03/bestPractices4/screen.do [26] (accessed on 2 November 2021). 

Based on the collected data, we developed variables for further statistical investigation. To adjust for the differences in infection risks among municipalities, we considered COVID-19 cases per 100,000 population, which was used as a response variable. Population density was adopted to partially account for the human-to-human transmission mechanism [28,29,30]. In this study, we discretized population density based on quartiles, taking the first quartile (Q1) as the reference category, so as to examine its non-linear influences on COVID-19 diffusion. Covariates associated with age structure were also considered. We chose relative proportions of the selected age cohorts (i.e., aged 10–19 and 20–29) since these groups appeared to be more active, potentially leading to wider and more rapid transmission of COVID-19. Furthermore, we employed two composite local safety level indices—Living safety and Infectious diseases—to account for the regional preparedness against external threats. For model-based formal assessments, we recategorized each measure into three levels: Good, Normal, and Bad from the original five levels of local safety level indices. To be more specific, Level 1 is assigned to the top 10% of the 226 municipalities followed by Level 2 with next 25%. Levels 3 and 4 account for next 30% and next 25%, respectively. The bottom 10% is attributed to Level 5 [26]. In this study, the class Good is defined by combining the Levels 1 and 2, Normal is Level 3, and Levels 4 and 5 are classified as Bad. 

### 2.3. Methodology

In explaining the spatiotemporal dynamics of monthly infections, we adopted a flexible Bayesian hierarchical approach with the Generalized Linear Mixed Models. This allowed us to account for the latent influences stemming from spatial arrangements of administrative units and temporal dependency along with the fixed effects by covariates. Following previous attempts in the disease surveillance domain [31,32,33,34,35,36,37], our analytic framework centers upon a Bayesian disease mapping methodology, whose major focus is the estimation of the relative risk of disease occurrence within a small area.

A systemic breakdown of the relative risks was considered in this study. We decomposed the unobserved risk surface of the disease into fixed (i.e., mean structure) and random (i.e., residuals including measurement errors, spatial correlation, and temporal dependency) components. Stated formally, the first level of hierarchy that deals with a data model is given by:(1)$$y_{s,\;t\;}\sim \;Normal\left(\mu_{s,\;t\;},\;\tau_{\varepsilon }^{-1}\right)$$
where the response variable, $y_{s,\;t\;}$, is the logarithmic transformation of the monthly accumulated number of COVID-19 cases per 100,000 detected in each municipality, *s*, on a month, *t*, from August 2020 to June 2021. We assumed that it followed a Gaussian (Normal) distribution with a mean, $\mu_{s,\;t\;}$, and a variance (an inverse of the precision, i.e.,$\;\tau_{\varepsilon }^{-1}=\sigma_{\varepsilon }^{2}$, related to measurement errors). 

The second level of hierarchy centers on the process that generated the data model. The process (i.e., mean structure), $\mu_{s,\;t\;}$, is characterized by the linear combination of fixed and latent random effects: (2)$$\begin{array}{c} \mu_{s,\;t\;}=\beta_{0}+\overset{p}{\underset{j=1}{\sum }}\beta_{j}X_{s,t,j}+v_{s}+u_{s}+\gamma_{t}\\ v_{s}\;\sim \;Normal\left(0,\tau_{v}^{-1}\;\right),\;u_{s}|u_{-s}\;\sim \;Normal\left(m_{s}^{-1}\sum_{k\in \partial_{s}}u_{k},\;{\left(m_{s}\tau_{u}\right)}^{-1}\right)\\ \gamma_{t}|\gamma_{t-1}\;\sim \;Normal\left(\gamma_{t-1},\;\tau_{t}^{-1}\right) \end{array}$$
where $\beta_{0\;}$is an intercept that explains an overall residual trend and $\beta_{j\;}$denotes *j*-th coefficient associated with the *j*-th covariate $X_{s,t,j\;}$assumed to be influential on COVID-19 diffusion risks. The convolution/Besag–York–Mollié (BYM) model, which is a sum of the intrinsic conditional autoregressive (ICAR) prior for $u_{s}$ and unstructured random effect, $v_{s}$, is widely adopted to account for spatial correlation [38]. Regarding the ICAR structure, $u_{-s}$ is the vector of the spatially structured latent effects excluding $u_{s}$ (i.e., $u_{-s}={\left(u_{1},\;u_{2},\;\cdots ,\;u_{s-1},\;u_{s+1},\;\cdots ,\;u_{76},\;u_{77}\right)}^{T}$) and $\partial_{s}$ refers to the set of neighboring regions for municipality, *s*, that allows us to define $m_{s}$, the number of neighbors of the *s*-th geographical unit [39,40]. This formulation indicates that for any *s*, the conditional expectation of $u_{s}$ is a weighted mean of its neighboring municipalities, with those being closer to *s* in some sense will have a higher contribution to that mean. 

Further, in this study we exploited a modified BYM model (BYM2) and the weakly informative penalized complexity (PC) proposed by Simpson et al. [41], which allowed us to overcome several limitations of the BYM model (for details, please refer to [41,42,43]). As shown in Equation (3), the BYM2 specification is characterized by a mixing parameter (ϕ) and a marginal precision parameter ($\tau_{s}$). The cofactor $u_{s}^{*}$ is the scaled ICAR component and $v_{s}$ corresponds to the unstructured random effect defined in Equation (2). This scaled re-parametrization of the conventional BYM model ensures we can verify the relative contribution of each component on the overall spatial correlation. Meanwhile, to address temporally structured random effects, ($\gamma_{t}$), we adopted a nonparametric random walk of order 1 (RW1) process [39,40,44,45].
(3)$$\begin{array}{c} \mu_{s,\;t\;}=\beta_{0}+\overset{p}{\underset{j=1}{\sum }}\beta_{j}X_{s,t,j}+\theta_{s}+\gamma_{t}\\ \theta_{s}=\frac{1}{\sqrt{\tau_{s}}}\left(v_{s}\sqrt{1-\phi }+u_{s}^{*}\sqrt{\phi }\right)\\ \phi \in \left[0,\;1\right]\\ \gamma_{t}|\gamma_{t-1}\;\sim \;Normal\left(\gamma_{t-1},\;\tau_{t}^{-1}\right) \end{array}$$

The third level of hierarchy focuses on (hyper)parameters related to the process. We assume the following:(4)$$\begin{array}{c} \beta_{0}\;\sim \;Normal\;\left(0,\;10^{4}\right)\\ \beta_{j}\;\sim \;Normal\;\left(0,\;10^{4}\right)\;for\;j=1,\;2,\;3,\;\cdots ,\;p-1,\;p\\ Prob\left(\left(1/\sqrt{\tau_{s}}\right)>(0.5/0.31)\right)=0.01\\ Prob\left(\phi <0.5\right)=2/3\\ Prob\left(\left(1/\sqrt{\tau_{t}}\right)>1\right)=0.01\\ Prob\left(\left(1/\sqrt{\tau_{\varepsilon }}\right)>1\right)=0.01\\ s=1,\;2,\;\cdots ,\;77;\;t=1,\;2,\;\cdots ,\;11 \end{array}$$
where the priors of $\tau_{s}$, $\tau_{t}$, $\tau_{\varepsilon },\;$and ϕ are weakly informative of the Penalized Complexity (PC) priors as suggested by Riebler et al. [43]. Using the probability statement such that $Prob\left(\left(1/\sqrt{\tau_{s}}\right)>Lower\;bound\right)=\alpha$ and $Prob\left(\phi <Upper\;bound\right)=\alpha$, the PC priors are defined as above [17,41,44,46].

Based on this understanding, we present the following models to examine the latent influences of space and time on the spatiotemporal dynamics of COVID-19 after accounting for the fixed effects. Model 1 solely considers the fixed effects due to covariates, while Models 2 and 3 incorporate additional random effects (e.g., spatially and temporally structured influences) into Model 1, allowing more flexibility in a modeling framework. Specifically, we define the risks, $\mu_{s,\;t\;}$, as: (5)$$\begin{array}{c} Model\;1:\;\beta_{0}+\beta_{1}Age1019_{s}+\beta_{2}Age2019_{s}+factor\left(PopdenQ_{s,t}\right)+factor\left(Safety_{s}\right)+factor\left(Disease_{s}\right)\\ Model\;2:\;\beta_{0}+\beta_{1}Age1019_{s}+\beta_{2}Age2019_{s}+factor\left(PopdenQ_{s,t}\right)+factor\left(Safety_{s}\right)+factor\left(Disease_{s}\right)+\gamma_{t}\\ Model\;3:\;\beta_{0}+\beta_{1}Age1019_{s}+\beta_{2}Age2019_{s}+factor\left(PopdenQ_{s,t}\right)+factor\left(Safety_{s}\right)+factor\left(Disease_{s}\right)+\theta_{s}+\gamma_{t} \end{array}$$
where $PopdenQ_{s,t}$ refers to quartiles of population density with respect to a municipality, *s* and a month, *t*. Two local level safety indices,$\;Safety_{s}$ and$\;Disease_{s}$, are also included in the statistical analysis.

## 3. Results

This study relied upon the monthly aggregates of COVID-19 confirmed cases of 77 municipalities from 1 August 2020 to 30 June 2021 (77 districts × 11 months = 845 observations). For each observation, we further adjusted the localized differences in transmission risks by using the number of COVID-19 cases per 100,000 population. To implement the suggested Bayesian models, we utilized R-INLA [47], which is an efficient alternative to the conventional Markov chain Monte Carlo (MCMC) simulation for sophisticated Bayesian models. 

Table 2 presents the outcomes of our selected models based on the goodness-of-fit measures, the deviance information criterion (DIC), and the widely applicable information criterion (WAIC) [48,49]. It is worth noting that models with a smaller DIC (or WAIC) are preferable. For example, when we add a nonparametric temporal trend by the RW1 process on top of Model 1 (i.e., Model 2), both DIC and WAIC values are substantially reduced. This indicates that temporally structured latent effects can account for unexplained variation on residuals. We then verify that Model 3 outperforms Models 1 and 2 since we can observe a substantial drop of both DIC and WAIC values in Model 3, justifying our adoption of random effects for an optimal model selection. Hence, the discussion of the analysis outcomes is based on Model 3.

Covariates pertinent to regional demographic structure appear to be unimportant, as the 90% credible intervals for Age cohort 10–19 and Age cohort 20–29 contain zero (Table 2 and Figure 4). Meanwhile, population density is positively correlated with COVID-19 transmission risk. The estimated posterior mean of parameters such as population density Q2–Q4 suggests that in reference to Q1, if a certain municipality belongs to Q2–Q4, it is more likely to have an elevated risk. Moreover, local safety level indices have shown mixed outcomes. We do not find any statistical evidence for Living safety variables due to the large variability in their posterior distributions, whereas the covariate, Infectious disease Good, showed a negative association with the risks compared to the reference category, Infectious disease Bad. Put differently, a municipality with good level of preparedness for infectious disease can have a reduction in risk. We also present Figure 4 for visual aids that help with the understanding of the relative influences of each covariate on transmission risk.

Moreover, our analysis showed different types of unobserved heterogeneity in explaining the random fluctuation of the COVID-19 diffusion risks over the SCA. The posterior mean of the mixing parameter (ϕ) of the BYM2 model is estimated to be 0.673. This indicates that the marginal contribution of the scaled spatially structured random component is 82% (i.e., $\sqrt{0.673}=0.8204$), which is greater than that of its unstructured (independent) counterpart, revealing strong spatial dependency. Regarding temporal correlation, we employed an RW1 prior, finding that the influence was relatively large because of the smaller precision of hyperparameters (see Table 2
*random effects*). This is also depicted in Figure 5, which shows the posterior distribution of the standard deviation of each latent effect. The values on the X-axis of the top-right and bottom panels in Figure 5 suggest that temporally structured latent effects largely account for the overall variability of the residual relative risks. 

## 4. Discussion

Considering these findings, we present Figure 6, which is based on the estimated posterior mean of COVID-19 risks ($\mu_{s,\;t})$ and exhibits the spatiotemporal evolution of the disease across municipalities over time. A color gradient is utilized to reveal the relative differences of transmission risks. The redder the color, the more vulnerable the district is. Conversely, the bluer it is, the safer. Here, we set a half-closed interval (3, 3.5] as a reference of risks since the overall average value of our dataset (calculated to be 3.14) falls in this interval. 

We argue that a map-based representation developed from the spatial and temporal smoothing approach allows us to depict the dynamics of COVID-19 infection. The top panels of Figure 6 reveal that during the second wave (the first four months from August to November 2020), the risks appear to be relatively lower, and the patterns of diffusion are likely to be spatially correlated with their neighboring locations. Interestingly, several districts in Seoul show relatively higher than others in the SCA for these periods, revealing spatial dependency. Meanwhile, we can identify abrupt changes in the risks with the start of the third wave in December 2020, which persist through June 2021 (refer to the Middle and Bottom panels of Figure 6). Elevated risks over the SCA in December 2020 are noticeable, with the highest intensity in many districts of Seoul (highlighted in darker red). Spatial correlation appears to continue over time since districts of elevated and/or reduced risks are clustered together in Seoul and Gyeonggi province, respectively. In other words, the pattern of the evolution of the risks for each district shows that it is spatially and temporally structured rather than random. 

In sum, we verify that districts with similar relative risks tend to be located together, implying spatial dependency. The spatial pattern also changes with respect to temporal shifts, which further reveals a temporal correlation. This suggests that accounting for random effects in the modeling process gives rise to an improved understanding of the risks of COVID-19 spreading across the SCA.

From a different perspective, we calculated the probabilities of risk estimates that were greater than a given threshold, the so-called exceedance probability [37,44,46]. These probabilities are useful to evaluate unusual elevation of risks. In this study, we set a threshold value of 3.5 because it is greater than the average of the whole study region and period (i.e., 3.14). It is worth noting that the antilog of a threshold value 3.5 is equal to 33.16, which is the monthly aggregates of confirmed cases per 100,000 population. We plotted Figure 7 exhibiting the probability that a municipality exceeds the threshold value, 3.5, in all maps. The top four panels of Figure 7 suggest that none of the municipalities in the SCA were estimated to have excess risks during the second wave of the pandemic. Conversely, central municipalities in the SCA (most likely ones in Seoul) have shown higher probabilities (close to one) over the third wave. We can infer that those colored blue (i.e., having probabilities greater than or equal to 0.9) very likely have risks in excess of 3.5. 

Successful identification of regions of excess risk (i.e., hot spots) confirms that there exists both spatial and temporal correlation as municipalities tend to be clustered in certain regions and the patterns persist over time. This can guide regional safety planners and policy makers as to where they should focus their efforts to mitigate the transmission risks of COVID-19.

There are several limitations that need to be discussed in this study. Although we have collected daily confirmed cases of COVID-19, we converted them into the monthly accumulated counts for each municipality. This aggregation may not fully reflect the actual localized diffusion process over the SCA, let alone temporal lags of COVID-19 cases. As a solution to this issue, in a future research endeavor we will rely upon Guglielmi et al. [50], whose approach (i.e., partial-differential-equation-based delay differential equations) can account for dynamics such as time delays due to incubation periods and memory effects in infectious diseases. 

Regarding covariates such as age structure and local safety level indices, we failed to incorporate a temporal dimension due to data unavailability. We utilized population density as a proxy for human-to-human transmission mechanisms, but population movement and interaction may be more relevant [51,52]. A plethora of studies have already shown the evidence of pollution and environmental impacts on COVID-19 diffusion (see recent systemic review papers [53,54,55]). However, this study focuses on social environments, and thus did not include pollution and climates.

Last but not least, our study does not factor into the modeling framework the effects of vaccination and policy measures, or nonpharmaceutical interventions, including social distancing, mandatory mask wearing, travel restrictions, active contact tracing, and tracking of the infected. Several studies suggest that vaccination reduces transmission, although the efficacy of COVID-19 vaccines differs between pre-Delta and Delta variants [56,57,58]. Thus it is desirable to incorporate the influence of vaccination into the modeling process. Unfortunately, there exists no publicly available timeseries vaccination data at the local municipality level in South Korea. The government only reports the cumulative total of daily nationwide vaccine doses, starting from June 2021 [59]. As a result, we failed to obtain a balanced panel dataset of vaccination, and consequently our spatiotemporal models do not include covariates specific to vaccination.

## 5. Conclusions

This research attempted to shed light on the space-time evolution of COVID-19 spread in the capital regions of South Korea during the second and third waves of the pandemic. In particular, we examined the following topics: (1) the effects of the local safety level indices on COVID-19 risks; (2) the role of unobserved heterogeneity, such as spatial and temporal correlation; (3) the location of clusters for further policy intervention.

To this end, we adopted a novel Bayesian approach as an analytic operational framework, which allowed us to incorporate latent influences from the spatial alignment of municipalities and their serial (temporal) correlation on top of the fixed effects by covariates. We specifically considered variables focused on sociodemographic features and localized preparedness for safety. Additionally, the impact of population density on COVID-19 transmission was assumed to be nonlinear, leading us to utilize quartiles. 

Our study reveals that a municipality with higher population density is likely to have an elevated infection risk, whereas one with good preparedness for infectious disease would tend to have a reduction in risk. Furthermore, we verified that addressing spatial and temporal correlation in the modeling framework significantly improves the performance and explanatory power, justifying our adoption of latent effects. Temporally structured effects are found to have relatively larger influence on the overall variability of residuals as compared to spatial correlation and measurement errors. Based on these findings, we present the dynamic evolution of COVID-19 across the SCA, which helps us verify unique patterns of the disease as well as regions of elevated risk for further policy intervention. Finally, we believe that our research contributes to improving the understanding of the pandemic in South Korea by providing insights from a different perspective, and thus complementing other works in this domain.

## Figures and Tables

**Figure 1 ijerph-19-00824-f001:**
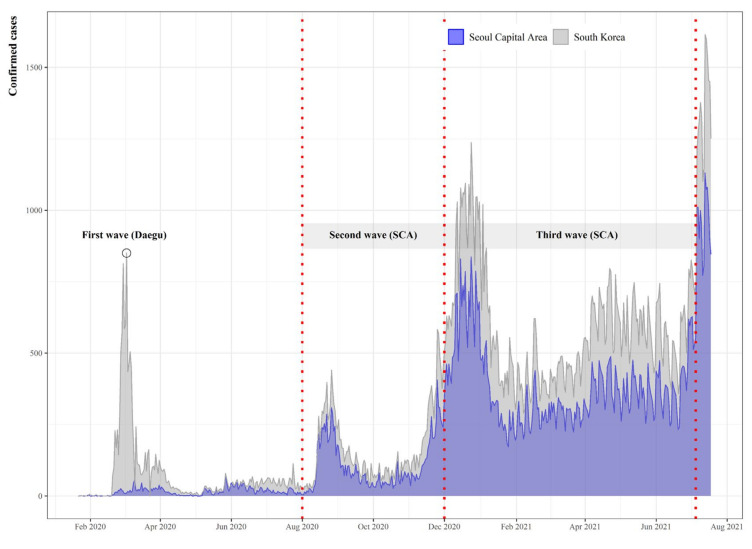
Daily confirmed cases of COVID-19.

**Figure 2 ijerph-19-00824-f002:**
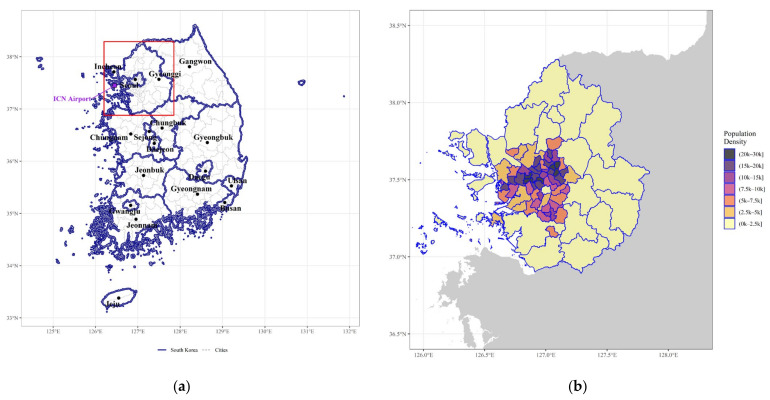
(**a**) Map of South Korea; (**b**) Population density of the Seoul capital area.

**Figure 3 ijerph-19-00824-f003:**
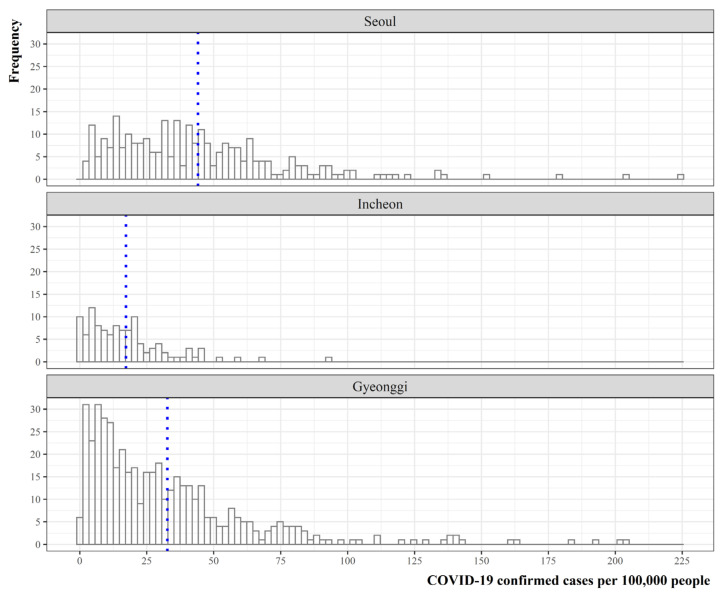
Distribution of COVID-19 confirmed cases (per 100,000 people). Note: the blue dotted vertical lines indicate the average number of confirmed cases for each province.

**Figure 4 ijerph-19-00824-f004:**
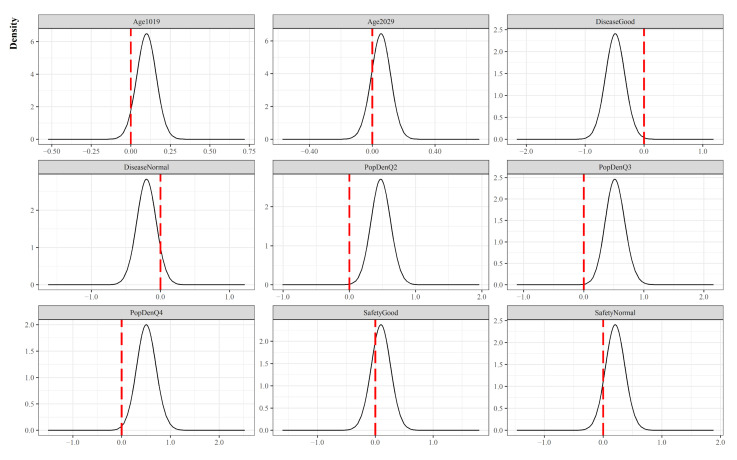
Marginal posterior distribution of covariates (Fixed effects).

**Figure 5 ijerph-19-00824-f005:**
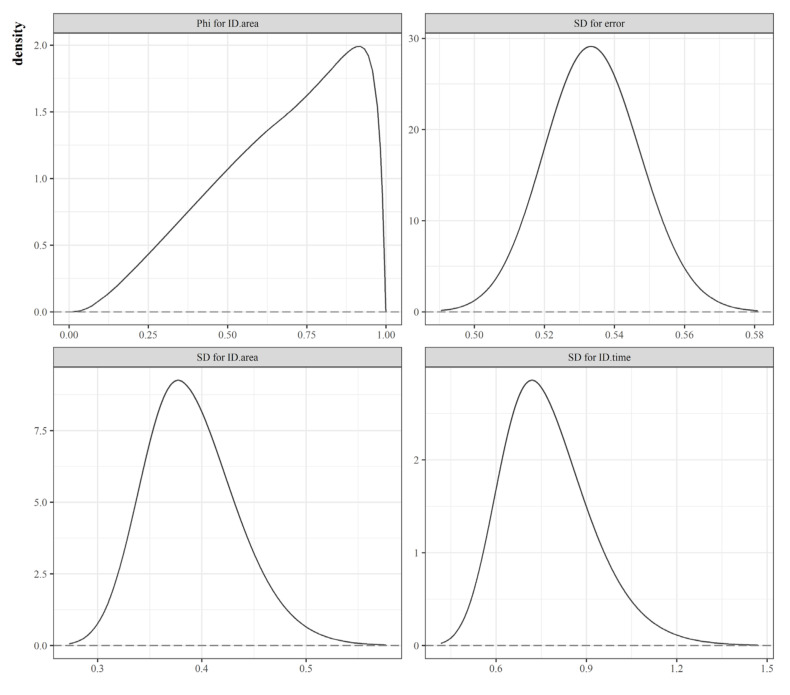
Marginal posterior distribution of hyperparameters.

**Figure 6 ijerph-19-00824-f006:**
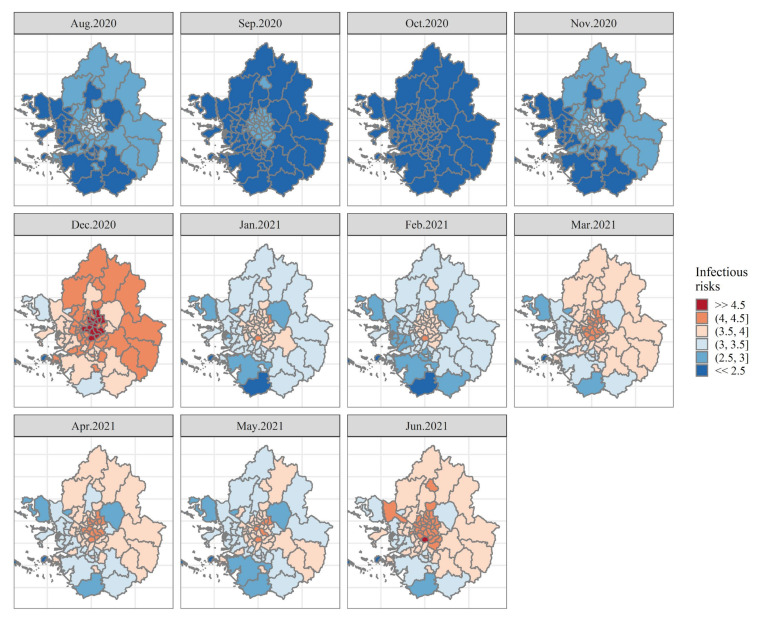
Spatiotemporal diffusion risks of COVID-19 during the second and third waves of the pandemic.

**Figure 7 ijerph-19-00824-f007:**
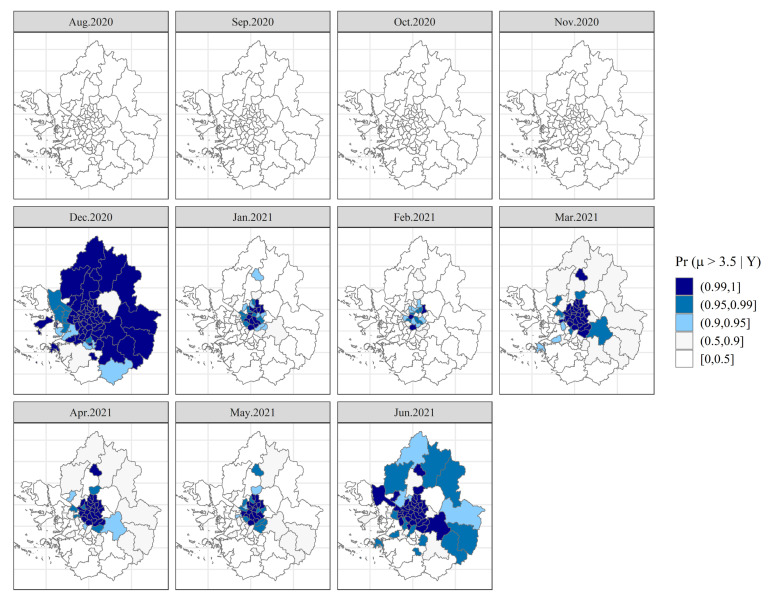
Exceedance probabilities of the dynamics of COVID-19 transmission risks.

**Table 1 ijerph-19-00824-t001:** Data in detail.

Data attribute	Description	Temporal Dimension	Sources
COVID-19 cases	Number of confirmed cases	Daily (1 August 2020~30 June 2021)	Each municipality
Population	Population count	Monthly (August 2020~June 2021)	KOSIS ^1^
Population density	Population per km^2^	Monthly (August 2020~June 2021)	KOSIS ^1^
Age 10–19	Percent of population aged 10–19	Census 2020	KOSIS ^1^
Age 20–29	Percent of population aged 20–29	Census 2020	KOSIS ^1^
Local safety level index ■ Living safety ■ Infectious diseases	Index value (Levels 1–5)	2019	MOIS ^2^
Spatial data	Geographical boundaries	Census boundary 2020	SGIS ^3^

^1^ Korean Statistical Information Service (https://kosis.kr/eng/ accessed on 2 November 2021) [25]. ^2^ Ministry of the Interior and Safety (https://www.mois.go.kr/frt/sub/a06/b10/safetyIndex/screen.do accessed on 2 November 2021) [26]. ^3^ Statistical Geographic Information Service (https://sgis.kostat.go.kr/view/index accessed on 2 November 2021) [27].

**Table 2 ijerph-19-00824-t002:** Regression results.

	Dependent Variable: The Natural Log of Monthly Aggregates of COVID-19 per 100,000					
	Model 1 ^3^		Model 2 ^3^		Model 3 ^3^	
	Mean (S.D.)	90% C.I.	Mean (S.D.)	90% C.I.	Mean (S.D.)	90% C.I.
* Fixed effects *						
Intercept	3.083 (0.118)	(2.889, 3.277)	3.106 (0.079)	(2.975, 3.236)	2.984 (0.169)	(2.702, 3.259)
Age cohort 10–19	0.076 (0.046)	(0.001, 0.151)	0.076 (0.031)	(0.025, 0.126)	0.100 (0.062)	(−0.003, 0.202)
Age cohort 20–29	0.057 (0.047)	(−0.021, 0.135)	0.067 (0.032)	(0.015, 0.119)	0.055 (0.063)	(−0.049, 0.157)
Population density Q2 ^1^	0.427 (0.118)	(0.233, 0.621)	0.354 (0.079)	(0.224, 0.485)	0.476 (0.148)	(0.234, 0.723)
Population density Q3	0.312 (0.117)	(0.120, 0.504)	0.285 (0.078)	(0.156, 0.414)	0.524 (0.164)	(0.260, 0.799)
Population density Q4	0.400 (0.136)	(0.176, 0.624)	0.356 (0.091)	(0.206, 0.506)	0.510 (0.202)	(0.182, 0.845)
Living safety Normal ^2^	0.314 (0.119)	(0.118, 0.510)	0.317 (0.080)	(0.186, 0.448)	0.207 (0.168)	(−0.068, 0.484)
Living safety Good	0.269 (0.115)	(0.080, 0.458)	0.285 (0.077)	(0.159, 0.412)	0.098 (0.170)	(−0.182, 0.377)
Infectious disease Normal ^2^	−0.267 (0.104)	(−0.438, −0.096)	−0.265 (0.070)	(−0.380, −0.151)	−0.201 (0.143)	(−0.435, 0.034)
Infectious disease Good	−0.730 (0.112)	(−0.915, −0.544)	−0.723 (0.075)	(−0.847, −0.599)	−0.489 (0.167)	(−0.764, −0.213)
* Random effects *						
$\tau_{\varepsilon \;}$(measurement error)	1.07 (0.052)	(0.986, 1.16)	2.38 (0.117)	(2.19, 2.57)	3.515 (0.181)	(3.225, 3.821)
$\tau_{t}$ (precision of RW1)			1.89 (0.7385)	(0.91, 3.26)	1.862 (0.722)	(0.907, 3.207)
$\tau_{s}$ (marginal precision)					6.818 (1.547)	(4.513, 9.565)
ϕ (mixing parameter)					0.673 (0.216)	(0.269, 0.958)
* Goodness of fit measure *						
DIC	2360.61		1693.89		1418.29	
WAIC	2360.87		1696.58		1423.78	
Marginal log-Likelihood	−1238.04		−930.13		−775.52	

^1^ The first quartile of population density is used as the baseline. ^2^ For the two local safety level indices, we regard Bad as the reference category. ^3^ For comparison purposes, we provide Model 1 (No structured latent effects), Model 2 (with temporally structured effect), and Model 3 (having both spatial and temporal effects as suggested in equation (5)).

## Data Availability

This study exploits publicly available datasets from various sources as given in Table 1. For research purposes, a series of data manipulation processes has been conducted. Hence, the data presented in this study will be available on reasonable request from the corresponding authors.

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
