# Peer review of "Spatiotemporal Associations between Local Safety Level Index and COVID-19 Infection Risks across Capital Regions in South Korea"

_ijerph, 2022, doi:10.3390/ijerph19020824_

Round 1
Reviewer 1 Report
Dear Authors,
thank you for your contribution on an up to date COVID-19 topic. Its good to support the understanding of the pandemic as well as influencing factors.
From my perspective the contribution is well structured and clearly written. Nevertheless, I have to note down few little annotations. On line 142 to 144 it seems unclear how the two paragraphs are connected or the paragraph on line 143 ends very abrupt. Secondly on line 320 and 321 you have changes in the referencing style maybe you can change this to the applied numbered style to help the reader find the references. Last, I like it very much that you have formulated research questions (lines 85 to 92) what I would prefer if you resume the research questions in the conclusions and make the answering clearer for the reader.
Best
Reviewer 2 Report
In my opinion the paper is bad written. The authors do not account for the used methodology correctly (e.g. the spatial component is not accounted for) and do not account for the literature in this topic (e.g
Blangiardo et al. (2020) Estimating weekly excess mortality at sub-national level in Italy during the COVID-19 pandemic. PLoS One. ; 15(10): e0240286.
D’Angelo, Abbruzzo, Adelfio, (2021) Spatio-Temporal Spread Pattern of COVID-19 in Italy. Mathematics 2021, 9, 2454)
In my opinion the paper ca not be published in this current form
I
Reviewer 3 Report
See attached.

Round 2
Reviewer 2 Report
You have accounted for my suggestions and it is now improved
Reviewer 3 Report
The authors have responded to my concerns and I now recommend the paper for publication